# Safety Built Right in: Exploring the Occupational Health and Safety Potential of BIM-Based Platforms throughout the Building Lifecycle

Madeleine Hoeft [1] and Catherine Trask [2,*]

1    Department of Real Estate and Construction Management, School of Architecture and Built Environment, KTH Royal Institute of Technology, 11428 Stockholm, Sweden; hoeft@kth.se
2    Ergonomics Division, School of Engineering Sciences in Chemistry, Biotechnology, & Health, KTH Royal Institute of Technology, 14152 Huddinge, Sweden
*    Correspondence: ctrask@kth.se

**Abstract:** This article investigates the opportunities of using digital building platforms based on Building Information Modelling (BIM) to increase occupational health and safety (OHS) in building design, construction, operation and deconstruction. The data collection followed a mixed-method approach with a systematic mapping review and focus group discussions with industry practitioners from the Swedish construction and real estate industry. Use cases were identified from both venues, as were prevailing barriers, potential facilitators, best practices and future applications. The findings highlight OHS potentials of digital building platforms for Rule-Based Checking and Design Validation, Team Building and Communication, Site Layout and Task Planning, Real-Time Monitoring, Equipment and Temporary Structures, Robotic Task Performance and Learning and Documentation. A set of principles is proposed to promote a higher degree of lifecycle and stakeholder integration: (1) technology, (2) data and information, (3) business and organization, (4) people and communication and (5) industry structure and governance aspects.

**Keywords:** occupational health and safety; digital twin; building information modelling; building life cycle; construction safety; design for safety; construction management; facility management

## 1. Introduction

The building sector has one of the highest rates of accidents and fatal injuries each year both in greenfield construction and maintenance works. In 2020, the construction industry had the largest proportion of people reporting a workplace accident in the European Union: 4.2% compared to 2.4% for all industries combined [1]. Most commonly, accidents concern falling, slipping or interactions with machinery. In addition, factors such as stress and long-term harmful work postures further put workers' occupational health and safety (OHS) at risk [2].

While the need for safety measures is generally recognized, risk assessments are often performed by individual trades and in an ad-hoc manner. In common industry practice, the ultimate liability for accidents happening on a construction site is borne by the constructor organization, even if those accidents could have been avoided by earlier design adjustments or evaluations [3]. Frequently, risk assessment approaches are inconsistent across projects and rely on the tacit knowledge and experience of individuals [4] As a consequence, information is not passed beyond discipline silos to all affected project parties. This refers to both vertical integration along the supply chain and horizontal integration along the lifecycle of a building [5,6]. On a vertical level, collaboration with workers, cross-trade training for a broadened safety understanding and adaptive safety to reduce paperwork are encouraged [7]. Throughout the building lifecycle, the notion of lifecycle safety has been introduced [8] to "reflect safety concerns in all phases of the facility's lifecycle including

programming, detailed design, construction, operations and maintenance, retrofit and decommissioning" (p. 2). In this lifecycle context, the concepts of Prevention through Design (PtD) and Design for Safety (DfS) are deemed especially impactful [9,10], given that a large share of construction accidents is related to design.

As manual, 2D-based safety planning has significant shortcomings and is prone to human error [11], recent research has focused on exploring safety technologies to support risk management. For example, emerging construction safety technologies have been clustered into applications for: project safety design and planning, visualization and image processing, project monitoring, information management and Internet of Things (IoT), automation and robotic systems, accident prevention and structure evaluation [12]. It is notable that most of the solutions are linked to (or based on) Building Information Modelling (BIM) platforms [9,13,14]. This enables more comprehensive representations; for example, hazard identification, 4D scheduling or automated safety code checking [10]. Additional applications in the operation and maintenance phase include the use of BIM tools for fire safety management and the identification of safety attributes during repair activities [5]. BIM has also been explored to help in the transfer of building system information from the design and construction to the operation stage to minimize safety risks [15]. Beyond those areas, information management and maintenance safety evaluation in the design stage have been highlighted as future research areas [16]. It should be noted that more recently advancements have been made beyond the pure semantic representation of building components and systems. Here, the concept of a Digital Twin aims at "a more holistic socio-technical and process-oriented characterization of the complex artefacts involved by leveraging the synchronicity of the cyber-physical bi-directional data flows", through the integration of BIM data with real-time data from IoT sensors or artificial intelligence (AI) [17]. However, research on the technology potential for increased OHS is to a large extent limited to the planning and construction phases, while other stages have been neglected [10,18].

As collaborative work methods have become more popular, opportunities have emerged for promoting safety throughout the building lifecycle in the context of an integrated, platform-based collaboration process. To facilitate this, there is a need to summarize and analyze the most recent research on BIM and operational health and safety (OHS). The successful implementation of digital OHS tools requires mapping not only the potential use cases (in terms of safety hazards and proposed solutions) within each lifecycle stage, but also the corresponding barriers and success factors when attempting to implement BIM methodology for OHS purposes ("BIM for OHS") in real life project settings.

Existing studies of BIM methodology have focused mainly on environmental and economic performance evaluations throughout the lifecycle [19–21]. Many existing reviews on the use cases do not specifically focus on safety implications beyond acknowledging generic benefits across the project lifecycle and stakeholder dimensions [22–24] or were performed in the context of another industry, e.g., manufacturing [25]. Available reviews of the OHS applications of BIM methodology tend to provide a cursory summary of findings, with a focus on bibliometric/scientometric analyses and reliance on article meta-data [10,18] or a provide a broad overview of OHS-related BIM opportunities without stratifying by lifecycle stage and hazard type [26].

The resultant gap in the understanding of barriers and facilitators in relation to the successful implementation of BIM for OHS has been acknowledged [26]. The single identified review that did examine success factors for BIM implementation was not specific to OHS, and also did not stratify by lifecycle stage [24]. In addition, many of the existing reviews are based on studies published more than five years ago [10,23,24,26], and there is a need to update the state-of-knowledge with the substantial number of recent developments. Clearly, there is a gap in the knowledge base about the characteristics and enabling factors of a more integrated, BIM-based safety management.

In addition to a current summary of the literature, there is a need to understand how BIM for OHS opportunities are reflected in current practice. Although there are a few studies that investigate the implementation of BIM methodology for real-life OHS

applications [27,28], most of the literature base proposes processes, applications, plug-ins and opportunities with no reported implementation in project settings. In the rare instances where industry feedback for a specific solution is reported, research settings are heavily skewed towards U.S. and Chinese contexts, with little consideration of European and specifically Scandinavian perspectives. There is research indicating that BIM methodology is used in Sweden [29,30], indicating that there is an opportunity there to apply BIM methodology to improve OHS outcomes. However, the lack of published BIM implementation research for OHS in the Swedish context requires a qualitative exploratory approach that investigates the what, how and why of current industry practices directly from industry practitioners.

This paper aims to investigate how BIM-based digital platforms can be used to support OHS management. It investigates the potentials to leverage automation capabilities and platform structures for a closer collaboration between different stakeholders throughout the building lifecycle. Enablers and prevailing challenges will be identified to guide future actions for technology-based accident prevention in research and practice. To fill the previously identified research gap, the following questions are addressed:

1.　What are the opportunities for lifecycle OHS management with a BIM-based digital platform, as described by the peer-reviewed scientific literature?
2.　What characterizes current BIM for OHS practices in the Swedish context?

The findings of this paper set BIM methodology-based safety technology tools in the context of holistic lifecycle safety approaches. The paper will also shed light on the current state of the Swedish building industry and the common challenges encountered in practice to highlight the need for further technology development and management training. A set of principles is derived to guide this transformation and inform stakeholder decision-making and actions.

## 2. Materials and Methods

The study design for this paper was a mixed-method approach, incorporating both synthesis of literature sources and qualitative focus group discussions with key informants. As described by Pluye and Hong, mixed methods combines "the strengths of quantitative and qualitative methods and to compensate for their respective limitations" [31].

### 2.1. Literature Review

This review was structured as a systematic mapping review, as described by Grant et al [32]. Consistent with their Search, Appraisal, Synthesis and Analysis (SALSA) framework [32], the aim of this mapping analysis was to characterize research streams addressing OHS strategies linked to BIM methodology. The findings subsequently informed the workshop questions for key informants in the Swedish building industry.

#### 2.1.1. Search and Screening Methods

A search was conducted in two main electronic published databases from: Scopus and Web of Science on 15 June 2021. Based on preliminary investigations into this literature, date limits were set for records published in 2010 and later. Full-text, peer-reviewed records in English were included. The main search terms included three conceptual groups of synonyms for "occupational health and safety", "building lifecycle stages" and "BIM". Synonyms within concept groups were combined with "OR", all conceptual groups were combined with "AND" (see Table 1 for a full list of search terms). Note that the asterisk (*) is used as a 'wildcard' operator or a placeholder which will return matches with different word endings. For example, Ergonom* will return 'ergonomics', 'ergonomist', 'ergonomic', and 'ergonomical'.

**Table 1.** Literature review search terms; terms within conceptual groups were combined with "OR" and all three conceptual groups were combined with "AND".

| Concept 1 Occupational Health and Safety | Concept 2 Building Lifecycle Stages | Concept 3 Building Information Modelling |
|---|---|---|
| Occupational health Occupational safety Safety management Accident prevention Injury prevention Ergonom* | Design Construction Facilit* management Demolition | BIM Building information modelling |

### 2.1.2. Inclusion and Exclusion Criteria

The review included peer-reviewed journal papers published in the English language, of all study designs (e.g., cross sectional and longitudinal designs, case studies and study protocols) as well as quantitative, qualitative or mixed methodologies. Given the anticipated early state of research in this area, no exclusions were made based on study quality.

Eligible records reported on the actual or future application of digital BIM with occupational health as a primary or secondary goal or benefit, within the context of building lifecycle, including design, construction, commission, operation and maintenance, renovation and deconstruction. Records investigating the building lifecycle as related to single-site, long-term (i.e., "permanent") commercial and residential buildings designed for continuous occupancy were included; infrastructure cases (e.g., railway, roads, nuclear plants) were excluded, as were temporary structures and small seasonal accommodations.

The scope of Occupational Health and Safety was considered to include the recognition, evaluation and control of hazards in relation to occupational exposure thresholds and related health effects. For example, noise levels approaching 90dB were included, but acoustic features related to speech intelligibility and annoyance were not. This interpretation of hazards extended to those onsite workers who are directly related with stages in the lifecycle; all types of construction workers, facility management and operation workers were included. To evaluate the direction and maturity of the field, primary research, reviews and editorials were included. Conference papers and study protocols that primarily described planned future research were excluded.

Identified records were screened for adherence to the inclusion criteria independently by two reviewers (MH, CT) at the title stage with any discrepancies resolved through discussion and consensus; inclusion and exclusion criteria were refined as needed based on this discussion. At subsequent abstract and full-text stages, these refined criteria were applied by at least one reviewer, with any ambiguous records reserved for discussion.

### 2.1.3. Data Extraction and Analysis

As there was no exclusion based on study quality features, there was no formal quality assessment. Since the studies differed considerably in their designs and characteristics, it was not considered appropriate to conduct a meta-analysis. Instead, the analysis took the form of a narrative synthesis of the main findings. The analysis of the journal papers was guided by the data categories and study characteristics outlined in Table 2. The category definitions summarized both general paper characteristics and specific use case details. An overview of this assessment for all included papers can be found in Annex Table 1. The main literature results took the form of a narrative synthesis of the main findings from primary research articles. Reviews were excluded from the analysis of primary articles, though many were cited in the discussion section to interpret the results.

**Table 2.** Data extraction categories and description of extracted information.

| Category | Description of Extracted Information | |
|---|---|---|
| Basic study information | - | Country of study |
| | - | Type of building |
| Solution characteristics | - | Data sources |
| | - | BIM applications |
| | - | Linked technologies |
| | - | Type of hazard addressed |
| | - | Type of solution |
| Stakeholder integration | - | Responsibilities for solution |
| | - | Beneficiaries of solution |
| Lifecycle integration | - | Lifecycle stages |
| | - | Links between stages |
| Impact and adoption | - | Facilitating factors for adoption |
| | - | Barriers/weaknesses of solution |

## *2.2. Industry Perspectives*

A series of workshops and interviews were held to address the following research question: What are the experiences of current professionals in Sweden with using BIM for OHS practices in a Swedish context, both in terms of existing (technological) solutions and in terms of barriers and facilitators in the implementation? Three 2-h web-conference sessions were conducted using focus group methodology adapted to the online format. These were supplemented by three one-on-one interviews using the same questions and prompts.

### 2.2.1. Recruitment and Sampling

Since the goal of the workshops and interviews was to investigate the perspectives of industry professionals in the intersections of OHS and BIM, the workshops were promoted via email, LinkedIn and Twitter using the professional networks of the authors and the stakeholder advisory group. Participants were encouraged to contribute to snowball sampling by sharing the invitation within their networks. When recruiting for the interviews, experience with using BIM methodology or digital twins at the specific lifecycle stage in question was an explicit eligibility criterion. Use of BIM is not legally mandated in Sweden, so the participants represented a specialized group of industry professionals. The result was a purposive sample recruited from professionals who work with BIM-based platforms in four stages of the asset lifecycle: design, construction, operation and deconstruction (see Table 3). One of the aims was a balanced gender distribution, resulting in 33% female and 66% male participants. Although a variety of professions participated (e.g., architects, engineers, site managers, (sub-)contractors, digitalization professionals, facility managers), there was group homogeneity from shared experience within a stage of the asset lifecycle.

Workshops were held for the design phase (3 participants), construction (6 participants) and operation phases (3 participants). To accommodate participants' schedules, two supplementary interviews were held for the operation phase. Two different workshop times were scheduled and promoted for the demolition phase but were cancelled due to low registration; ultimately one interview was conducted for this phase.

**Table 3.** Professional roles of interview and focus group participants within each lifecycle stage.

| Lifecycle Stage | Profession (n) of Workshop Participants | Profession (n) of Interview Participants |
|---|---|---|
| Planning and Design | Architect (1) Structural Engineer (1) Design Manager (1) | - |
| Construction | Architect (1) CEO for Sub-contractor (1) Construction Engineer (1) BIM Coordinator/Project Manager (2) Head of Design Team (1) | - |
| Operation and Maintenance | Architect (1) Health and Safety Manager (1) Property Manager (1) | Software Developer (1) Property Manager (1) |
| Demolition and Reconstruction | - | Sustainability Consultant (1) |

2.2.2. Facilitation

Using a phenomenological framework and an inductive approach [33], questions and prompts aimed to elicit participants' direct experiences and perceptions related to their current or planned implementation of BIM methodology. The question topics regarding use cases, facilitators and barriers were selected to be comparable to the main themes of the literature review. In addition, a question about potential upcoming applications was added to ensure a transparent distinction between currently implemented real-world use cases and industry practitioners' prognoses, conjecture or hypotheses. Feedback from the construction industry steering group related to this project was also considered to ensure the questions' relevance and applicability to industry practice. The main question topics used for both workshops and interviews are shown in Table 4.

**Table 4.** Outline of the open-ended questions used in the industry workshop discussions.

| | Question Topics |
|---|---|
| 1 | Please describe your use cases for BIM to enhance health and safety. What actors were involved and what were the main information sources? |
| 2 | Which factors supported the implementation of the use cases? |
| 3 | What are the main challenges to implementing BIM for safety benefits? |
| 4 | How could these barriers be overcome? |
| 5 | How (else) could BIM and digital twins be used for safety in future applications? |

Workshops and interviews were led by the authors using facilitation best practices as previously published [34,35], with adaptations for the web conference format [36]. For example, in the absence of in-person body language, during workshops the facilitators endeavored to hear from all participants by using a round-robin format. Following introductions and orientation, workshops opened with a warm-up activity using Mentimeter, a web-based group polling platform that allows for open-ended responses and displays results as a word cloud or set of anonymous quotes. Participants were invited to share their main safety priorities, motivation for implementing BIM and main safety applications for BIM. This was followed by group discussion in a round-robin format with broad, open-ended questions. Interviews followed the same process, with the warm-up questions posed and answered verbally, and follow-up questions to the individual participant replacing the round-robin format. Notes were made on participant responses and reactions. Workshop and interviews were also recorded with the permission of the attendees; all participants provided informed consent prior to participating.

### 2.2.3. Data Analysis and Synthesis

Analysis of the workshop responses followed an inductive approach using qualitative content analysis as described by Graneheim et al. [37]. Analysis was performed by both members of the research team. One of the team members performing analysis had specific training and professional expertise concerning BIM methodology and building lifecycles (MH), while the other had corresponding training and expertise in occupational health and safety (CT). Workshop notes were first open-coded then grouped into relational categories and themes within each lifecycle phase. Direct quotations from participants were selected to illustrate each aspect of the findings and to demonstrate that the analytic interpretations were rooted in the data. The literature review and industry perspective data were compared and combined into a single synthesized visual representation of categories and themes.

## 3. Results

### 3.1. Summary of Literature Review

After removing duplicates, the literature search yielded 272 individual records. A full-text review left 79 papers that were considered related based on the defined selection criteria. A total of 51 of them were non-review articles that met the inclusion criteria and were retained for full data extraction. Figure 1 shows the results of the screening process.

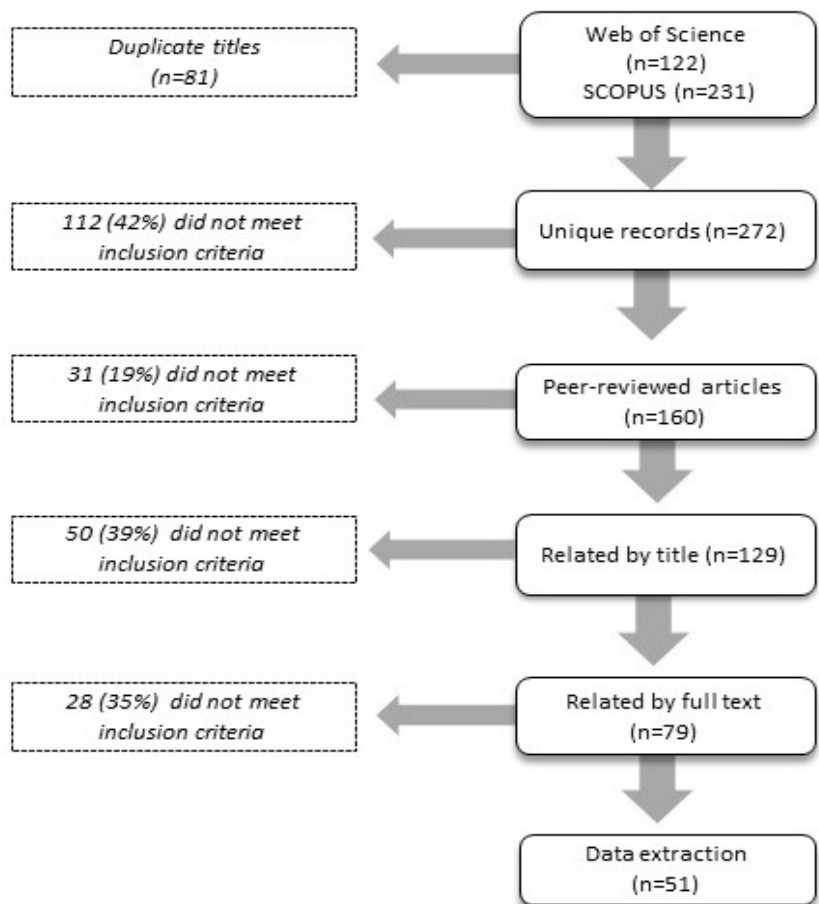

**Figure 1.** PRISMA diagram outlining the results of the literature search, screening and extraction stages.

The majority of the 51 primary research articles were published in 2015 (10), 2016 (7) and 2020 (11). A total of 25.5% (n = 13) of the articles included consideration of the design phase, 60.8% (n = 31) of the construction planning phase, 47.1% (n = 24) of the

construction execution phase and 13.7% (n = 7) of operations phase. No papers were identified addressing hazards in deconstruction or demolition. (Note that some articles considered more than one phase). Figure 2 summarizes the distribution of the main hazards and solution types across the lifecycle stages.

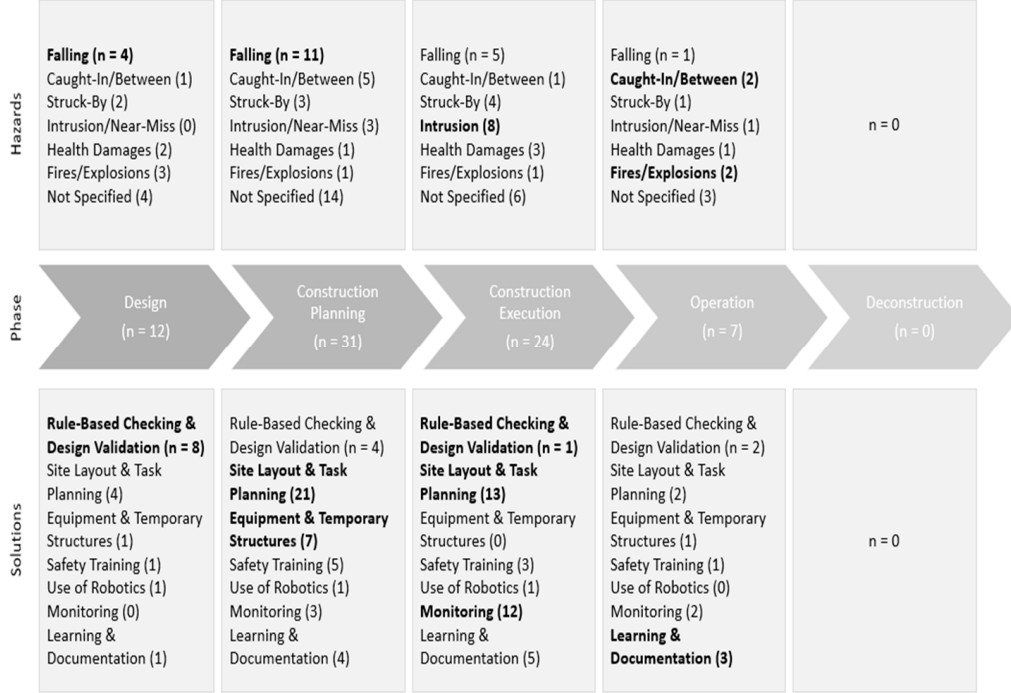

**Figure 2.** Hazards and BIM-based solution types with associated lifecycle phases as described in the published literature.

The use cases discussed in the literature covered a variety of solution types which can be broadly grouped into seven categories: Rule-Based Checking and Design Validation, Site Layout and Task Planning (workspace planning), Safety Training (simulation, gamification), Real-Time Monitoring (surveillance, tracking and notifications), Equipment and Temporary Structures (scaffolding, protective personal equipment), Robotic Task Performance and Learning and Documentation (knowledge management, reporting, decision-making). A full tabulation of the extracted data from the 51 records included in the review can be accessed as Supplementary Materials (Table S1).

### 3.1.1. Use Cases
Design

Research publications focusing on applications of BIM methodology for safety enhancements during the design stage mostly presented Design for Safety approaches for rule-based checking and validation. While the primary objective of those papers was to increase the safety of construction workers, a few solutions also aimed to improve OHS aspects for facility managers and building occupants. The tools were typically intended to support architects and engineers, for example, through automated design review and risk assessment systems [4,38,39], and simulations to optimize design choices with respect to emergency situations such as fire evacuation [40]. In addition, the use of knowledge databases was explored in an attempt to bridge the gap between construction and operation knowledge and early-stage architectural design [16,41,42]. Finally, using BIM to design for robotic construction can be considered to reduce human exposure to hazardous work [43]. The data used in the safety tools came from historical data aggregated on an industry level [38] or building level [44], safety guidelines and national building codes [40], as well as professional knowledge gathered in interviews or from project documentation [45,46].

If information was passed across lifecycle stages, it was mainly to navigate robotic equipment [43] and to validate or document as-built conditions [5,45]. The majority of the reported solutions (63%) were based on 3D BIM models and did not report the use of any other technologies to support the BIM methodology-based digital environment for safety purposes.

Construction Planning

Tools in construction planning often address the site layout and task planning (67.7%), as well as the use of equipment and temporary structures (22.6%) to prevent falls and other accidents during the work execution. As such, proposed solutions included, for instance, automated safety planning [47–49], the identification of blind spots and danger zones on site [50] as well as workspace planning and training for the interaction with machinery and the assembly of heavy elements [28]. Thereby, 4D BIM (including the link to a construction schedule), was employed in 54.8% of the articles to visualize dynamic sequences and to better account for risks related to changing site layouts and building conditions. The safety managers were described as the central actors to employ the solutions either as the sole target user or to coordinate the use with actors from the design or construction execution phase for feedback and planning validation. Fifteen of the thirty-one articles drew on professional experience as input data, historical data were used in seven cases, while safety guidelines such as OSHA were used in twelve proposed solutions. In addition, individual worker characteristics and work sequence descriptions were taken into account to plan and evaluate construction workspaces with the help of Virtual Reality (VR) [28,51].

Construction Execution

Safety during the construction execution phase was, first and foremost, addressed by BIM methodology-based solutions that enabled real-time monitoring, notifications, visualizations and warnings, as well as safety compliance checking. They aimed to increase the degree of process automation for safety tasks [51–53], prevent exposure to harmful environmental conditions [54] and improve on-site safety communication, for example through real-time reporting of unsafe conditions [55]. In addition, the aim was to improve the availability of assembly information [38]. The main hazards discussed in the literature were unauthorized intrusions and near-misses as well as falling and struck-by accidents during interactions with heavy objects. Data sources were mostly on-site real-time data (50%) or industry safety guidelines (29.2%). To date, a number of technologies have been linked to the digital platforms, including RFID tags, GPS and Bluetooth beacons for localization, drones for image production, robots for automated installation, block chain for secure credentialing of materials and approvals and IoT sensors to monitor environmental conditions such as dust or heat.

Operation and Maintenance

During the operation stage, the main hazards investigated concerned emergencies such as fires and explosions, as well as maintenance work in confined spaces [56,57]. Emphasis was also placed on the applicability of digital platforms for documentation purposes to provide guidance for inspection and maintenance tasks [5,58]. Data sources primarily included input from experienced professionals and past project reports (57.1%) as well as industry guidelines (42.9%). Since there were no dynamic changes in the building layout, all solutions in the operation stage were based on 3D BIM-based platforms. In terms of additional technologies, several solutions carried out the integration of these platforms with IoT sensors, sometimes explicitly introducing the notion of a digital building twin [56].

3.1.2. Barriers

The aforementioned use cases spanned a variety of purposes and intended benefits, but the included articles rarely reported measurements or assessments that quantified the impact of using BIM-based platforms for safety. Outside of technical simulations and

hypotheticals, the barriers and facilitators of real-world implementations were based on researchers' interviews with users and hence represent subjective perspectives. Overall, the barriers reported in the academic literature can be grouped into five categories: (1) technology, (2) data and information, (3) business and organization, (4), industry structure and governance and (5) people and communication.

With regards to technology, data and information, barriers to the adoption of BIM for OHS throughout the building lifecycle concerned the technological immaturity of the solutions in terms of hazard detection and data processing capabilities. For instance, the tools fell short on accounting for interdependencies between different risk factors in the complex construction setting [45,46,59]. Commonly, solutions for falling or collisions were presented, neglecting other hazard sources such as electricity. Moreover, complementary infrastructures such as RFID antennas or a strong WiFi-network would be needed to enable, for example, cloud-based information sharing and real-time monitoring. However, this infrastructure is seldom present at today's construction sites, is itself immature (e.g., limited sensor accuracy), or requires a lot of resources and space to install [57,60]. Other barriers to BIM-based solutions include the limited user-friendliness of the software interfaces [54] and limitations of the input dataset to identify risks in the first place [39,52,61]. This latter limitation often stems from a lack of detailed accident statistics on a corporate or industry level, especially with regards to individual trades or small and medium enterprises (SMEs) [62]. In addition, the building model has to be modelled very accurately in order to automate safety-related tasks. This makes the design phase very time-consuming, especially if the information is static and does not update automatically based on identified hazards, simulations, and schedule changes [39,63]. Human intervention is hence often needed to validate outcomes and ensure their quality [62,64], resulting in a high degree of manual work and overall lower automation levels.

In terms of business and organization aspects, the main barriers to using BIM-based platforms for safety include the lack of defined evaluation metrics and difficulties to account for e.g., behavior-based safety hazards. This limits the inclusion of safety criteria in traditional time- and cost-centered decisions [51,65]. Moreover, the need for organizational changes and the currently isolated nature of BIM methodology use cases can make it difficult to scale solutions to a profitable level [43]. Many safety applications require a high level of detail of the building models, which in turn could be leveraged, for example, for more prefabrication to justify the higher upfront resource needs. From an industry structure and governance perspective, the regulatory landscape reportedly provides insufficient support for the application of BIM for OHS due to the lack of standards [4,62] or difficulties to obtain approval for the use of onsite equipment such as drones [27].

In terms of people and communication factors, Choe and Leite claim that the lack of technical skills of construction and design staff is one barrier to employing the solutions at larger scale [47]. Few professionals in the industry possess the programming skills that would enable them to customize, for example, the integration of different tools such as BIM and VR [28]. In addition, integrity concerns were stated in the context of real-time monitoring [55] or the fear of job losses and resistance to the adoption of new technologies [57].

### 3.1.3. Facilitators

Next to the barriers, there were also a number of factors mentioned that support the development and implementation of BIM-based platforms for safety use cases. Most commonly, these included integrated project team structures fostering communication links between actors from several lifecycle phases and organizations [38,63], as well as a high level of software interoperability [54,66]. Regarding the latter, there is evidence that a workload reduction can be achieved in a single software environment as a unified system of integrated components. In this context, several papers reported the use of established BIM software such as Revit or Navisworks as a facilitating factor [15,48], while others emphasized the importance of open standards such as IFC [40], or—beyond a sole BIM focus—the introduction of standards for Common Data Environments (CDE).

To create a more collaborative, web-based environment, cloud platform technologies as the backbone of the communication infrastructure have become increasingly popular [57,60,63]. For any of the solutions, a high degree of usability (i.e., intuitive handling, no programming knowledge requirements and ergonomic hardware) and accessibility on mobile devices (especially for fieldwork) is deemed crucial to easily retrieve safety-related information [13,28]. Other factors that can help to introduce a BIM methodology for lifecycle safety include regulatory obligations [48], the link to other use cases such as prefabrication [56] and a high level of detail about the underlying circumstances, for instance, the occupancy characteristics of a building in the case of fire simulations [44]. Finally, solutions with low implementation costs and an innovation mind-set within the project organization(s) are needed to scale the adoption in the industry [51].

Figure 3 provides an overview of all barriers and facilitators found in the literature review including their categorization into (1) technology [T, visualized in Figure 3 as blue boxes], (2) data and information [D, in green], (3) business and organization [B, in orange], (4), industry structure and governance [I, in yellow] and (5) people and communication aspects [P, in purple].

**Figure 3.** Barriers and facilitators as described in the published articles included in the literature review.

*3.2. Summary of Industry Perspectives*

Throughout the entire lifecycle, the workshop participants described perceiving the use of BIM-based platforms for safety purposes in the Swedish building industry as currently "exploited to a small extent" according to polls conducted during the sessions. Irrespective of the lifecycle stage, they are mostly adopted to realize cost and time savings. However, there was a strong potential and need seen for leveraging the potential of digital, platform-based applications for safety, especially when it comes to long-term perspectives and the connection of silos in the building ecosystem.

### 3.2.1. Use Cases

Design

Existing industry use cases described in the design phase primarily fall into the areas of (1) Rule-based Checking and Design Validation, as well as (2) Team Building and Communication. Typically, these solutions in the design stage are based on 3D platforms (mostly architectural elements), sometimes combined with VR tools or game engines. For instance, the creation of virtual game environments was described as a way to explore the designs in a realistic simulation about 6–12 months before the actual completion of construction works. Feedback can be given much earlier to detect and prevent hazards in a timely manner, and this can be applied in both construction and maintenance scenarios. Joint virtual explorations help designers, construction managers and facility managers to communicate and identify spatial constraints and requirements at an early project stage:

> *"We like [the construction managers] to think ahead: while we design, we want them to look at safety aspects for raising the building during construction. [ ... ] Does it work the way it is designed?"*

Additional value stems from virtual walks in site areas which are typically inaccessible due to, e.g., ongoing nearby transport operations, to visualize and explore the site conditions and building dimensions. It was also reported that this gamification approach increased the team feeling and consequently the interaction and trust among the stakeholders, leading to higher perceived safety. Additionally, using BIM-based software in design meetings and visual risk assessments was reported to facilitate discussions and help to pass safety information from design to construction staff by including annotations about potential hazards and safety instructions directly in the models. Most often, the solutions described target design managers, architects, safety and construction managers. Stakeholders and information from the operation and deconstruction phase were considered less frequently to not at all.

Future use cases that were brought up by participants in the design workshop were related to the improvement of software and hardware functionalities as well as the data and information quality. Regarding technology, more automated risk assessments in tools such as Navisworks or Solibri and the integration of generative design in project workflows were perceived as useful to promote and structure the consideration of safety aspects at an early project stage. In addition, the further customization of the software interfaces was deemed important to allow for a smooth transition between different stakeholder perspectives (management, site managers, project leaders, etc.) and the related information displayed. With regards to data and information, the curation of industry databases and open information exchange based on common standards was suggested, exemplified by the idea of a "Github for the construction industry". Closely linked to this aspect was the notion of lifecycle learning, aiming to save the data relevant for safety risks and incorporate feedback from construction and operation into the design phase to increase data-based decision making and prevent hazards stemming from design choices.

> *"We should not just design good houses if we are lucky with the [project] group. There are sometimes groups that work well together, sometimes less and it is important that technology supports us in the future to find errors. Then people can still make a choice."*

Construction

On site, commonly mentioned hazards were falling from height, struck-by and dangerous environmental conditions such as dust and heat. To address them during the construction phase, BIM methodology-based solutions in the Swedish industry can be broadly clustered into applications for (1) Site Layout and Task Planning, (2) Real-Time Monitoring and (3) Learning and Documentation. While the majority are based on 3D platforms, some approaches feature connections to sensors, location tags and smart personal protection equipment or a scheduled integration for 4D BIM simulations.

Applications for site layout and task planning included the use of BIM methodology-based tools for safety workshop communication as well as the planning, evaluation and follow-up of work sequences including the corresponding placement of building materials. During workshops, BIM methodology was reported to be the basis for the dialogue with designers about assembly considerations and potential sequencing options. For instance, during the installation of MEP systems, daily logs and notes on the BIM-based platform helped the general contractor to track material and completed steps, perform tests on time and prevent potentially hazardous interferences of trades on site. Moreover, clearly planning and marking material locations in BIM increases transparency for all trades involved, reduces stress and optimizes the workers' paths to avoid collisions and spatial constraints. An example mentioned was the assembly of inner walls. The information used from the building model included the size of the pre-cut boards and their intended location in the building. The delivery packages were then grouped and placed on-site the day before the assembly without any disturbance of simultaneous ventilation and electricity works. With the use of 4D platforms, additional opportunities arise for the evaluation and visualization of spatial constraints in a more dynamic manner. However, while considered a promising option by several workshop participants, 4D BIM methodology was not reported to be currently implemented for construction sequencing beyond initial trials.

Real-time monitoring use cases included the use of wearable personal equipment such as smart helmets that help to both prevent accidents and send real-time notifications in case of a worker's fall or being struck. With the help of location markers, workers' positions are anonymously tracked and visualized in a digital building twin, which can support emergency evacuations and the monitoring of danger zones with entry restrictions. In addition, environmental sensors send real-time alerts if the air quality (e.g., humidity, dust, gases) or noise levels pose a threat to workers' health.

Finally, BIM methodology was reported to support learning and documentation efforts at inspections during and after the completion of construction works through as-built comparisons, annotations in the models and real-time sharing of information linked to the respective building locations. While it often starts with the project managers using BIM methodology, the greater safety benefits were seen from the integration of a variety of stakeholders, including cleaning personnel and subcontractors. The goal was described as:

> " ... *to have everyone's eyes both in the building model and in the real world on site and to see potential issues.*"

Concerning future use cases of BIM for OHS, workshop participants discussed a variety of potential developments. Although blockchain security has not been used to this point, it could be a useful facilitator to secure and certify data updates. As described by one informant, the data itself would become an important product particularly in digital twins:

> " ... *[the] allegiance to [the] building over time is greater than allegiance to one property owner*"

As another example, the increased use of BIM-based game environments or VR applications could bring additional stakeholders such as architects, owners and the general public into the project to raise awareness of spatial and temporary constraints during construction. As mentioned in the design stage, the importance of a lifecycle perspective on safety and a higher degree of customization with regards to the information displayed was also stressed in a construction phase (and operation) context. The extension of BIM model dimensions to 4D further enables a more dynamic visualization of spatial and temporal implications during constructions to minimize hazards from overcrowding and a lack of coordination. In addition, potentials were identified for more prefabrication and robotics to cut hazardous on-site work performed by humans. BIM methodology is essential here as the underlying information source for manufacturing and assembly. Next to that, leveraging artificial intelligence to handle the enormous amount of data generated over time and across projects will be essential in the long run. It was however also stressed that these transformations are likely to take time and require a clear "safety first" mind set:

*"We have to stop asking yes or no questions: If it is about using digital twins for safety, it can only have a yes answer"*

Operation

In operation, the use cases mentioned by industry practitioners can mainly be classified as solutions for learning and documentation. Since the building is completed at this point, all solutions are represented in 3D. During operation, more examples of the integration of various sensor data sources were mentioned than in construction, aiming for a shift from BIM to digital twins as a virtual duplicate of the complete asset. This includes, for example, the integration of checkpoints in the building to link check-up rounds and safety instructions to a digital twin. The underlying intention was to decouple information from single individuals and promote safety through shared information visualized in a 3D map environment. Knowledge about maintenance standards and work procedures can thus be made explicit, whereas it was previously typically trapped in software silos, on paper or in the head of individuals. Digital notations in the building model were mentioned to allow for additional remarks and follow-up tasks. Other use cases that are being explored but not yet widely implemented in Sweden at this point include the use of localization sensors to automatically filter safety information based on a worker's location. Over time, the aggregation of maintenance information to data-driven reports was suggested as a way to further promote safety by enabling more proactive operations and the timely replacement of worn-out equipment. Additionally, real-time monitoring was mentioned as a safety-related use case for digital building models. Here, sensor data linked to building locations can be used to visualize equipment breakdowns, including information about the necessary repairs. Moreover, it can trigger alerts about unhealthy air conditions to enable a timely response by the maintenance staff and/or the general public in the building.

Asked about the direction for the next five to ten years, potential was seen in the extended use of real-time data linked to building locations for more automated building system steering and control as well as for managerial decision-making. In this context, the threat of cybersecurity was mentioned, representing a new dimension of safety to be considered as buildings become smarter and more connected. Another aspect currently neglected is social safety in residential districts—a factor that was reported as not being linked to digital models yet despite being of high importance to residents. Moreover, the need for more integrated collaboration including the suppliers and other ecosystem actors was mentioned to promote safety through building platforms.

Deconstruction

At present, the use of BIM for OHS is very limited in deconstruction. In pilot projects, 4D simulations are performed by project managers to evaluate different scheduling scenarios and visualize work sequences for different stakeholders. With the increasing importance of circularity, potential synergies from the integration of lifecycle safety aspects into material databanks and a closer, optionally BIM methodology-based collaboration between circularity experts and safety managers were mentioned; however, no current use case was identified by the participants.

3.2.2. Barriers

Although linked to experiences with use cases in their respective lifecycle stages, the barriers described by industry professionals showed a lot of consistency across lifecycle stages. Therefore, the barriers to using BIM methodology for safety applications are presented here not within use cases or lifecycle stages, but according to thematic categories. Qualitative analysis of the workshops and interviews yielded five overarching categories of barriers: (1) technology; (2) data and information; (3) industry structure and governance; (4) business and organization; and (5) people and communication.

The theme of technology related to both lagging technology maturity and the limited scalability of solutions. Inconsistent platforms across firms and across lifecycle stages made

it difficult to realize the benefits of a common information source. For example, it was considered faster to adopt and implement new digital tools in the design stage than in the construction phase, so the latter lags behind. Concerns about limited scalability related to the time- and effort-intensive process of building models adequate to support safety efforts; this in turn led to limited coverage of assets. In the early stages of technology maturity and usability, firms experience the labor and cost without yet benefitting from easy access to shared data. Additionally, low usability reduced the ability to make use of new technologies; it can be difficult to find the appropriate safety information in a model when needed. Most software solutions do not offer built-in workflows that facilitate safety applications, requiring external consultants or staff with specialized programming knowledge.

Barriers within the data and information theme largely described issues with acquiring safety-related (e.g., safety measures during construction, maintenance instructions, accident statistics) data in a usable format that support transferability across platforms, particularly across stages in the lifecycle. Those collecting and supplying data inputs for BIM may not be working with safety specifications in mind. The responsibilities for safety may be spread among several outsourced consultants or sub-contractors, and the ultimate decision-making power for implementing safety processes may lie within a different life-cycle stage, or with an actor from a different firm. According to workshop participants, the lack of understanding about the information needs of each lifecycle stage, and the potential for that information to have a positive safety impact, is a major barrier. For example, without key inputs from the operation and construction phases, it is not clear what safety hazards should be highlighted at the design stage, and what type of data to pass forward to future stages. There remains a lack of clear requirement definitions to guide the development of model information inputs:

> "[OHS data for BIM] needs to become . . . an industry standard. I think we are going in that direction . . . but it is still like some of the companies have THEIR solution and other companies have THEIR solution . . . "

At the same time, they acknowledged a trade-off between strict and detailed requirements that would increase cost, vs. loose low-detail requirements insufficient to enhance safety. In addition, public clients stressed the perceived conflict between neutral tendering documents and the implementation of standardized routines in information management.

The industry structure and governance theme described barriers related to how the industry's existing frameworks and practices impacted the implementation of BIM for OHS. For example, particularly in construction, the industry is fragmented and decentralized, with diverse and local firms contributing to a finished project; this "ecosystem web" can make it difficult to adopt new technologies and platforms without standards and regulations enforcing consistency. With small margins and considerable uncertainty over multi-month projects, this is a setting with considerable unavoidable business risk and risk aversion in terms of early adoption. Perceived financial risks were described by a construction workshop participant as a disincentive to be an early adopter:

> "I want to test new things, but I don't want to be the first guy out."

This industry tendency towards risk aversion is linked to the theme of business and organization, which pertained to challenges in evaluating the impact of adopting BIM for OHS, and in particular quantifiable evidence of success. Without specific metrics, "better safety" or "better workflows" are vague and intangible goals that are not very enticing from a business perspective. When queried as to what type of evidence would be useful, one operation workshop participant described:

> "Cost and time, because that's how decisions are made at the top . . . "

The lack of links to cash flow combined with the potential for long (or unknown) returns on investment are a particular challenge within the temporary project context of construction stage. Business and organization barriers also involve prioritizing organizational functions, allocation (and siloing) of work and the lack of a "road map" or set process for integrating BIM methodology into workflows throughout the lifecycle stages.

*"There is a missing link: We need to use the BIM model all the way, which is not really the case in many projects today. ... When you can connect the whole value chain all the way to operations and also follow what is done and how it is done, then quality and safety can improve."*

Technology adoption is typically driven by core productivity goals such as cost and time savings. Safety is rarely a primary organizational priority, and since high safety performance is not typically listed as a contract specification by clients, it is often one of many ancillary regulatory requirements competing for attention. An organization's work allocations can present a barrier when the workers handling digitalization and safety are in separate departments, resulting in a silo effect that inhibits the sharing of information and the development of updated work processes that meet several goals (e.g., production and safety goals).

The people and communication theme concerns barriers with human resources, and the need to further develop communication, knowledge and leadership relative to BIM for OHS. Communication gaps include how data are used and what is needed at each stage, and knowledge gaps often reflect a "myopia" within a lifecycle stage or professional role. Similarly, professionals with an artistic focus in architectural design and working in the design phase may not know many details about the construction process, building logistics and time spans; this limits the degree to which they can contribute relevant safety-related data. A lack of big-picture management capacity to identify hazardous situations in advance via coordination between highly skilled individuals was also described within the theme of "people". This would require a prioritization of leadership that incorporates generalized skills and knowledge over specific focus on one profession or stage:

*"Everyone is working with a certain part of the project and sometimes they have excellent tools in the bigger projects [for calculating, follow-ups, model viewing], but the core construction skills and having the oversight to manage a project well is sometimes getting lost. That stops us from taking bigger steps. It is happening, but it is still slow."*

### 3.2.3. Facilitators and Best Practices

Facilitators for the adoption of BIM for OHS were proposed as potential tactics to mitigate the barriers named by informants; the best practices were current methods or strategies that had already been tested. The best practices and facilitators described by informants largely fit into the same five overarching themes: (1) technology; (2) data and information; (3) industry structure and governance; (4) business and organization; and (5) people and communication.

Best practices in technology related to realism, accessibility and interoperability. Success was seen with technology that provided highly realistic representations to facilitate an intuitive understanding of 3D and 4D models. Current accessibility best practices also included portable access via cloud networks on mobile devices, with suggested facilitators being availability to a large number of stakeholders to create a multi-professional team with an understanding of the digital environment and representations. For example, wider accessibility would be facilitated by technology that allows a large number of stakeholders to contribute to the development and updating of digital building models in construction and operation. User-centered interfaces should be tailored to provide the information that is needed for each user and to avoid information overload. Open-source platforms and standards could reduce friction between programs and platforms and allow for accelerated data usage and greater interoperability; ultimately this could contribute to greater collaboration and integration between firms, professions and lifecycle stages.

Informants described data and information best practices including flexibility, accessibility and information consistency. There has been success with flexible and accessible data structures that will facilitate the crowdsourcing of information (keeping models up to date) and enable transparent, shared access to a "single source of truth" which can be considered a current and trustworthy description of the asset. However, in terms of facilitators there remains a need for a clear industry-wide standardization of requirements and formats on

how to collect and link high-quality data. In order to facilitate data integrity over time, data structures should be linked by location attributes and should accommodate many sources and types of data into a database that can be shared and interpreted by multiple platforms or tools. Given that, ideally, data will be created and used across many firms and lifecycle stages, there is no competitive advantage to a single proprietary platform or data ontology. Rather, there was a stated need for agreement across lifecycles stages and firms on what type of information is needed, and what formats and structures are best to encode the information. Extending the utility of, for example, the building collaboration framework (BCF) could help achieve this, as described by a participant in the design workshop:

> " . . . there can be some rules and a language, for example added to BCF, to exchange that information. Maybe [safety hazard information] can also be connected to BCF, because there you can connect different views and models. This can be a way of transferring information from one program to another. We can use [multiple current BIM and DT platforms] . . . and everyone receives the same information and it is editable. BCF gives a lot of possibilities for this, [it] is not that far away".

Best practices in industry structure and governance include repositioning the legal value of BIM documentation. For example, elevating BIM models as the highest legal document governing assets throughout the lifecycle provides an additional incentive for BIM to become the "single course of truth" regarding the asset.

In terms of business and organization, best practices related to realizing the commercial potential of BIM methodology adoption, developing Key Performance Indicators (KPIs), and incorporating cross-functional work practices. Cross-functional integration has successfully spanned lifecycle stages, gathering input from stakeholders in other stages (even end-users and building occupants). As a current best practice, traditional metrics such as time and cost are being augmented by engagement metrics such as daily active users (DAUs) of the tool, and interactions with safety-related data as an indicator of safety management performance. Additional best practices include the development of new business models and incentives for early investment that can help counteract a culture of risk aversion.

> "The moment you can talk business about it, a lot more people will be interested to help, especially from the top."

Proactive use and tracking of data in earlier lifecycle stages can demonstrate when there are future savings. It has proven beneficial to start small with pilots, proof-of-concept and test beds to demonstrate profitability and then to scale up. Informants reported that orientation towards innovation has grown when (best) practices shift towards experimentation, allowing for quick iterations of "fail-and-learn", but this needs to be developed further through test beds, continuous benchmarking and sharing the results:

> "We don't take any risks at all in this business" . . . "You have to fail to learn"

Moreover, as investors and shareholders demand corporate actions on sustainability and ESG (environmental, social, governance) compliance, this is seen as another force to drive the implementation of platform-based, digital safety solutions.

Under the people and communication theme, BIM for OHS requires developing the workforce, specifically in terms of skill capacities, an innovation/digital mind set and cultivating support and leadership for BIM methodology-supported safety. In current best practice, workforce development is motivated by a shared vision of possibilities for safety supported by top management and operational staff, and by inclusive contributions from all stakeholders. This approach engenders a sense of pride; for example, in the operations phase there comes a sense of comprehensive stewardship and knowing a building inside and out, and being proactive in forestalling maintenance issues:

> "You want to have 100% control and you can in a better way. [ . . . ] People almost start to compete with each other to have better knowledge and information connected to the digital twin."

Mutual communication and sharing of best practices and success stories was reported as a current best practice, and also one that could be expanded further as a facilitator. There was great value seen in the promotion of both small-scale firms and high-profile "Hollywood projects" that have demonstrated the successful application of BIM for OHS; the result was that the benefits and impact of data sharing through BIM-based platforms became more widely understood. While there might be disincentives to sharing some types of best practices with competitors, informants described that this did not apply well to safety issues, since enhancing working life and attracting workers to the industry is an industry-wide issue, and not specific to any one firm. According to informants, best practice shows facilitation, support and prioritization from several stakeholders; when general contractors, top-level management and clients champion BIM for OHS, it helps promote the wider adoption of the technology. Although the implementation of BIM for OHS requires the work of many, it is currently driven by management and client demands, and in particular, demands from those who are responsible for the costs, delays and/or legal repercussions of a workplace accident. When leadership and clients understand the link between safety and project quality, their motivation for adopting BIM for OHS is increased:

> *"With intense planning and a better run project, safety will come or if you look from the other direction, if you are safety-focused, you will plan your projects better and then also lower the costs for the client in the end so the involved parties can make more money."*

Although current best practice involves developing a strong understanding of the types of data and communicating how to use it throughout the lifecycle, this needs to be further developed in order to exploit efficiencies and enhance safety. Informants proposed building the internal skills capacity to allow even small firms to go from relying on external consultants, data specialists and BIM specialists, to all actors contributing to and benefiting from BIM methodology. When properly facilitated, this capacity could grow and reach a critical mass, as described by one informant:

> *"The change will be seen in the moment where we don't need those specific roles any longer, when everybody understands what BIM means and how to use the tools we have. Like with the telephone—when it was invented, you needed a telephonist to help. When people can find information they need by themselves, the real change will happen."*

Figure 4 provides an overview of all barriers and facilitators brought up during the workshops, including their categorization into (1) technology [T, visualized in Figure 4 as blue boxes], (2) data and information [D, in green], (3) business and organization [B, in orange], (4), industry structure and governance [I, in yellow] and (5) people and communication aspects [P, in purple].

Barriers and Facilitators – Workshop Discussions

| Facilitators | Lifecycle Stage | Barriers |
|---|---|---|
| Frequent testbeds **B1** | | **D1** Limited data transferability |
| Client support **B2** | | **P1** Communication gaps |
| Lifecycle knowledge **P1** | Design | **B1** Lack of resources |
| Clear communication **P2** | | **B2** Lack of lifecycle perspective |
| Realistic virtual representations **T1** | | **B3** Time pressure |
| Regulatory predictability **I1** | | **B4** Low prioritization |
| Integrated project teams **B3** | | **P2** Lack of technical skills |
| Access for many stakeholders **T2** | Construction | **P3** Lack of project oversight |
| Early investments in safety **B4** | | **T1** Technological immaturity |
| Top-level prioritization **B5** | | **B5** Lack of implementation road map |
| Sense of pride and inclusion **P3** | | **B6** Lack of evaluation metrics |
| Public sharing of best practices **P4** | | **B7** Scattered responsibilities |
| Building of inhouse skill pool **P5** | Operation | **B8** Lack of client support |
| Clear transformation road map **B6** | | **I1** Regulatory boundaries |
| Adjustment of evaluation metrics **B7** | | **I2** Risk aversion of actors |
| Flexible database structures **D1** | | **I3** Different levels of adoption |
| Data crowdsourcing and sharing **D2** | | **I4** Fragmented industry nature |
| Information consistency **D3** | Deconstruction | **T2** Limited scalability of solutions |
| Lifecycle data perspective **D4** | | **D2** Lack of clear requirement definitions |
| Data transferability **D5** | | **D3** Lack of input data |

T = Technology, D = Data & Information, I = Industry Structures & Governance, B = Business & Organization; P = People & Communication

**Figure 4.** Barriers and facilitators as described in workshop discussions with industry professionals.

## 4. Discussion

### 4.1. Comparing the Literature Review and Focus Group Findings

In some cases, the findings from the literature review and focus groups were consistent and/or reinforced one another. As an example, several of the barriers/facilitators were consistent between the data sources related to technology (e.g., technological immaturity), data and information (e.g., data transferability) and governance (e.g., regulatory requirements) were consistent. In addition, the low number of demolition/reconstruction use cases reported in the literature seems to correspond with the low focus group participation within this lifecycle stage; it seems reasonable to assume that this reflects the slower adoption in current practice.

However, there was also a difference in that many of the implementation challenges and proposed success factors regarding human factors highlighted by the focus groups were not found (or were less prominent) in the literature. For example, focus group participants had insights on industry-level organizational culture such as client support and the re-prioritization of time and resource allocation, as well as on the need to develop BIM methodology-related skills and coordination roles. Although some of these same barriers were listed in the published research, in the literature they mainly arose from conjecture and prognosticating rather than directly from collected data. A very small proportion of published research studies have investigated the *implementation process* of BIM methodology; this may be why the barriers and facilitators gathered from the focus groups within the theme of "people and communication" were broader and more developed. This suggests an opportunity for implementation research that can evaluate and promote promising practices for the successful application of BIM for OHS goals.

### 4.2. Principles for the Successful Adoption of BIM for OHS

Findings from the industry discussions and systematic literature review revealed a number of potential use cases for BIM-based platforms to improve safety throughout the

building lifecycle. Whether a situation becomes hazardous or not is influenced by different causal hierarchy levels: originating circumstances (e.g., safety culture, client requirements or construction education), shaping factors (e.g., design specifications, worker skills or site constraints) and immediate accident circumstances (e.g., communication, material conditions or the temporary local climate) [67]. Drawing from these risk layers [67], this article identified core examples of how digital building models can support both technical and psychological safety factors by transforming: (1) the underlying industry structures (with regards to safety culture, fragmentation and client requirements); (2) risk management strategies in construction projects and building management (e.g., clearer design specifications, industrialized construction methods and immersive worker trainings); and (3) the immediate situations in which accidents can occur (e.g., digital communication, material conditions and local climate monitoring).

Implementing BIM methodology can be considered a prerequisite to the successful application of BIM for OHS use cases. However, given this article is specific to safety use cases, the summarized principles for the successful adoption of BIM (Figure 5) focus primarily on safety applications to guide this transformation and inform stakeholder decision-making and actions. The categorization of the principles is based on the earlier detailed discussion (for reference, see Figures 3 and 4).

| Technology | Focus on user-friendliness | Intuitive interfaces, customized display of safety-relevant information |
| | Modular stake-holder integration | Horizontal and vertical collaboration and safety communication among stakeholders |
| | Prioritize digital infrastructure | Early-stage investments in complementary infrastructure to enable real-time monitoring and safety communication |
| | Increase industrialization | Avoid hazardous tasks via more robotic task performance and off-site production |
| **Data & Information** | Establish a single source of truth | Use digital models as single information source throughout the lifecycle |
| | Plan digitally "all the way through" | Include all relevant information in digital model at planning stage for fewer ad-hoc adjustments on site |
| | Build a common data environment | Adopt database-driven structure with geometrical properties from BIM as one of several input sources |
| | Ensure platform flexibility | Use open interfaces to non-BIM tools (sensors, drones) to overcome BIM limitations/ enable parallel advancements |
| | Share safety information | Promote industry-wide sharing of safety information to enable learning at larger scale |
| **Industry & Governance** | Stronger academia & industry links | Ecosystem-level collaboration to coordinate education, research and industry practices on safety and digitalization |
| | Adjust regulatory requirements" | Make BIM part of regulatory safety decision-making on a(n inter)national level |
| | Establish standards | Industry-level standardization of data formats for low-friction sharing/transfer of safety-related information |
| | Scale solutions on industry level | Partner to create viable implementation cases and economically attractive markets for safety-focused solutions |
| **Business & Organization** | Democratize safety | Involve all stakeholders in safety discussions, reporting and learning with help of digital platforms & encouragement |
| | Adopt feasible metrics | Shift from pure cost to digitalization metrics like interaction frequency or engagement with technology |
| | Link safety and business goals | Identify how safety can be integrated to improve costs or schedule targets (avoid the "safety sidecar") |
| | Iterate in frequent testbeds | Test new solutions on a small scale first to allow more flexible implementation and quicker adjustments |
| | Scale in the project | Identify opportunities to leverage detailed modelling and documentation throughout building lifecycle |
| **People & Communication** | Cultivate a safety mind-set | Make safety a priority in communication and training across organizations, disciplines and hierarchy levels |
| | Cultivate a digital mind-set | Train staff to use digital tools to support their work, reduce stress and increase safety via high quality building data |
| | Communicate success stories | Share best practices and pilot results to raise awareness for safety potential and other benefits |
| | Respect workers' integrity | Where possible use anonymous tracking solutions to respect worker privacy and integrity and ensure buy-in |

**Figure 5.** Proposed guiding principles for implementing BIM for safety, developed from the literature and industry workshop findings combined.

### 4.3. Methodological Considerations and Significance of Results

The findings presented here complement previously published work by adding an industry perspective and extending the considerations over the entire building lifecycle, from early design to deconstruction. In setting the discussion about the potential of BIM methodology-based platforms for safety into a larger context of uninterrupted information flows towards digital twins, it moves beyond isolated use case scenarios as suggested by the authors of ref. [68].

By combining both academic reports and the experience of industry professionals, this paper also contributes to a better understanding of how BIM methodology could be applied to safety management, and what the barriers and facilitators would be for realizing that potential. Given the previously recognized lack of practical BIM methodology applications [69], this unique synthesis of data sources and the subsequent development of principles for future use are a major strength of the paper. Moreover, introducing a Swedish

industry perspective—a country known for its leading role in the promotion of worker safety and use of technology—represents an interesting complement to the work of the authors of ref. [70], who surveyed professionals in the United States.

The literature search included a limited date range, although since the first years of the search window did not contain any included papers, it seems unlikely that important research was missed using these date limits. It was noted that no papers were found for the demolition stage of the lifecycle; this could be the result of missing search terms needed to identify these papers. However, given that the industry workshops struggled to identify and recruit professionals in the demolition/deconstruction phase who had experience with BIM methodology, we rather consider our search to be an accurate reflection of the current level of BIM for OHS maturity in this lifecycle stage. Despite searching in international scientific publication indexing databases, we note that the included papers reflect a preponderance of research from Korea, USA and China. Although unlikely, it is possible that the English language search limits precluded important work published in other languages. The review extraction and synthesis approach used in this paper aligned best with that of a mapping review. However, the search and screening process involved the systematic application of broad search terms and clear inclusion and exclusion criteria; this rigorous methodology increases confidence that all relevant papers were included.

The phenomenological approach to the interviews was intended to temper the academic perspective on possible BIM methodology-based safety solutions with the lived experience of professionals currently practicing in this field. As with most qualitative interview studies, the samples were small and purposefully selected, which precludes both statistical analysis and any assumptions of the representativeness of the sample. Instead, the advantage of this approach is in the richness of the explanations and examples given, and the ability to generate hypotheses and frameworks for future studies to test. It should be noted that the workshops relate to the Swedish context, and that the sector's digital maturity, governance structures and the Swedish sociocultural milieu for both safety and technology are likely to have impacted the findings. While it was unfortunate not to be able to collect wide perspectives on BIM methodology within the demolition stage, we interpret this challenge to be a reflection of the lagging implementation of BIM in that stage.

Although the focus group findings fill some gaps in the current published research, the focus group methodology does not supply the data needed to make a reliable statement about similarities of Swedish practices to other countries, or differences between professions. Making those comparisons was not the original goal of this research project and would require further quantitative data collection (i.e., surveys) across professions in the Swedish industry or a comparison of practices in different countries. This is an important topic and it is hoped that the barriers, facilitators and priorities described by the focus group participants in the current study will be useful in developing survey items for future studies.

## 5. Conclusions

This article assessed how BIM-based digital platforms for operational health and safety during the building lifecycle, as guided by two questions: (1) "*What are the potentials for lifecycle OHS management with a BIM-based digital platform, as described by the peer-reviewed scientific literature?*" and (2) "*What characterizes current BIM-based OHS practices in a Swedish context?*". A mixed-method approach was chosen to investigate use cases, barriers and best practices in academic research and Swedish industry practice. Enablers and prevailing challenges were identified to guide future actions for technology-based accident prevention in research and practice.

BIM-related OHS solutions have the potential to improve the sustainability of both a productive construction workforce, and the healthcare systems which benefit from the reduced burden of caring for construction-related injury and illness. Such solutions are emerging in the fields of Rule-Based Checking and Design Validation (Design for Safety), Team Building and Communication, Site Layout and Task Planning, Real-Time Monitoring,

Equipment and Temporary Structures, Robotic Task Performance and Learning and Documentation. BIM methodology is hence not limited to visualizing building geometry but has significant usefulness as a data source in relation to a broader data environment from which stakeholders can retrieve unambiguous information catering to their needs at any point in the building lifecycle. As a consequence, trust in shared information, real-time hazard monitoring and availability of structured documentation contribute to promoting safety in buildings. Moreover, since occupational health and safety should be a joint effort of all stakeholders, the increased cross-functional teamwork and democratization of information management enabled by BIM methodology-based solutions can be of tremendous value for safety throughout the building lifecycle.

While the academic literature mostly reports shortcomings in terms of technological immaturity and missing complementary infrastructure on building sites, the Swedish real estate and construction industry described struggles with (technical) skill shortages among their staff and the low user-friendliness of existing solutions. Finding a balance between software expertise and building construction and maintenance experience will be key to benefitting from new technologies, making informed choices and not blindly trusting auto-generated results. In addition, few BIM use cases and investments today are motivated by safety as a key driver despite its high relevance to the industry, mostly due to the lack of adequate quantitative metrics.

To promote a higher degree of lifecycle and stakeholder integration and to overcome current limitations, this paper proposed a set of principles related to (1) technology, (2) data and information, (3) business and organization, (4) people and communication and (5) industry structure and governance aspects. These findings have implications for stakeholders in building design, construction, operations and deconstruction. They can help to define the next steps in the implementation of BIM use cases for safety, identify potential pitfalls and contribute to learning from successful pilot projects and approaches. To leverage BIM for OHS, it should not be viewed as an isolated task. Instead, it must become an integral part of BIM methodology and data management discussions linking various stakeholders throughout the lifecycle. It is hoped that enhanced digital maturity in combination with an understanding of the respective product and process impacts can prevent injury and illness and thereby enhance the health, sustainability and productivity of the construction and maintenance workforce. To fully leverage BIM for OHS, more research is needed to demonstrate the quantifiable benefits that justify potential higher initial costs and provide guidance in the implementation process. It will also be important to define standards and information requirements to make safety-related data an integral part of digital building models throughout the lifecycle.

**Supplementary Materials:** The following supporting information can be downloaded at: https://www.mdpi.com/article/10.3390/su14106104/s1; Table S1: Summarized literature review characteristics of the 51 included papers reporting BIM applications in occupational health and safety.

**Author Contributions:** Conceptualization, M.H. and C.T.; methodology, M.H. and C.T.; formal analysis, M.H. and C.T.; writing—original draft preparation M.H. and C.T.; writing—review and editing, M.H. and C.T.; visualization, MH and C.T. All authors have read and agreed to the published version of the manuscript.

**Funding:** This research was funded by AFA Försäkring, grant Dnr 170149.

**Institutional Review Board Statement:** Given the non-sensitive, non-personal nature of the data collected in this interview study, an institutional ethics review was not required within the Swedish Ethics framework.

**Informed Consent Statement:** Informed consent was obtained from all focus group participants involved in the study.

**Data Availability Statement:** Not applicable.

**Acknowledgments:** We gratefully acknowledge the review and feedback from Principal Investigator of the AFA Försäkring grant project, Professor Jörgen Eklund. We would also like to thank the industry stakeholders who contributed to this paper by participating in the focus groups.

**Conflicts of Interest:** The authors declare no conflict of interest.

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
