# Peer review of "Safety Built Right in: Exploring the Occupational Health and Safety Potential of BIM-Based Platforms throughout the Building Lifecycle"

_sustainability, doi:10.3390/su14106104_

Round 1

Reviewer 1 Report

The paper is well written and relevant. More engagement with debates in sustainability journal is recommended.

Author Response

Reviewer 1 Comments

The paper is well written and relevant. More engagement with debates in sustainability journal is recommended.

RESPONSE: Thank you for your comments. We have added more specific mention of sustainability issues related to our topic in the conclusion section.  This section now reads:

“BIM-related OHS solutions have the potential to improve the sustainability of both a productive construction workforce, and the healthcare systems which benefit from reduced burden of caring for construction-related injury and illness.  Such solutions are emerging in the fields of Rule-Based Checking & Design Validation (Design for Safety), Team Building & Communication, Site Layout & Task Planning, Real-Time Monitoring, Equipment & Temporary Structures, Robotic Task Performance, and Learning & Documentation.”

Reviewer 2 Report

Thank you for a very interesting paper. it is difficult to present this amount of detailed analysis in a readable graphic form but you have made a good effort to do so. The conclusions are succinct and carefully presented to avoid overstatement. Well done.

Please correct Line 265 - reference missing

Author Response

Reviewer 2 Comments

Thank you for a very interesting paper. it is difficult to present this amount of detailed analysis in a readable graphic form but you have made a good effort to do so. The conclusions are succinct and carefully presented to avoid overstatement. Well done.

Please correct Line 265 - reference missing

RESPONSE: Thank you for noticing this.  This is consistent with the comment from Reviewer 3.  We have fixed this by listing Figure 1 which is being referred to here; note that this is not a reference but rather a hyperlink within the document added by editorial staff. 

Reviewer 3 Report

  • In general, there is a lack of explanation of replicates and statistical methods used in the study. Furthermore, an explanation of why the authors did these various experiments should be provided.
  • Also, there are few explanations of the rationale for the study design.
  • The header of Table 4 should be on the same page as the table content.
  • L265,please check the content of the article again, the references are missing.
  • 3.1.1.1 in L574 has only one title, so is it unnecessary to use a fourth-level title?
  • Please check the format of Reference 5 again.
  • The references in the text are not listed in the order in which they appear.

Author Response

In general, there is a lack of explanation of replicates and statistical methods used in the study. Furthermore, an explanation of why the authors did these various experiments should be provided. Also, there are few explanations of the rationale for the study design.

RESPONSE: Thank you for this comment; it uncovers a need for more clarity in our methods.  We have not used an experimental study design, and thus do not have either ‘replicates’ or statistical methods.  We have used a mixed methods study design that combines (1) systematic mapping literature review and (2) qualitative focus groups.  We have adjusted the first sentence of the methods to better reflect this, and provide some rationale for a mixed methods approach.  It now reads:

“The study design for this paper is a mixed-method approach, incorporating both synthesis of literature sources and qualitative focus group discussions with key informants.”

In the methods sections that describe (1) the literature methods and (2) the focus groups, we have provided references and rationale for selecting those methods.  We have now also added a rationale and a reference for the use of mixed methods approach.  This section reads:

As described by Pluye and Hong, mixed methods combines “the strengths of quantitative and qualitative methods and to compensate for their respective limitations”.

The header of Table 4 should be on the same page as the table content.

RESPONSE: Good catch, we have changed this.

L265,please check the content of the article again, the references are missing.

RESPONSE:  Thank you for noticing this.  This is consistent with the comment from Reviewer 2.  We have fixed this by listing Figure 1 which is being referred to here; note that this is not a reference but rather a hyperlink within the document added by editorial staff. 

3.1.1.1 in L574 has only one title, so is it unnecessary to use a fourth-level title?

RESPONSE: Thank you for this remark. We fully agree that a section with only one title would not justify the use of a fourth-level title. However, the manuscript continues with 3.1.1.2 - 3.1.1.4, reflecting the different use cases per life cycle stage. For the sake of structure and readability we therefore chose to use fourth-level titles unless the editors prefer a different format.  

Please check the format of Reference 5 again. The references in the text are not listed in the order in which they appear.

RESPONSE:  Thank you for noticing this.  The reference list has now been updated, and we will work with the editorial team to make sure it is properly formatted. 

Round 2

Reviewer 3 Report

N/A